# Dynamic construction of refractive index-dependent vibrations using surface plasmon-phonon polaritons

Hong Zhou [1,2], Zhihao Ren [1,2], Dongxiao Li[1,2], Cheng Xu [1,2], Xiaojing Mu [3] ✉ & Chengkuo Lee [1,2,4,5] ✉

One of the fundamental hurdles in infrared spectroscopy is the failure of molecular identification when their infrared vibrational fingerprints overlap. Refractive index (RI) is another intrinsic property of molecules associated with electronic polarizability, but with limited contribution to molecular identification in mixed environments currently. Here, we investigate the coupling mode of localized surface plasmon and surface phonon polaritons for vibrational de-overlapping. The coupling mode is sensitive to the molecular refractive index, attributed to the RI-induced vibrational variations of surface phonon polaritons (SPhP) within the Reststrahlen band, referred to as RI-dependent SPhP vibrations. The RI-dependent SPhP vibrations are linked to molecular RI features. According to the deep-learning-augmented demonstration of bond-breaking-bond-making dynamic profiling in biological reaction, we substantiate that the RI-dependent SPhP vibrations effectively disentangle overlapping vibrational modes, achieving a 92% identification accuracy even for the strongly overlapping vibrational modes in the reaction. Our findings offer insights into the realm of light-matter interaction and provide a valuable toolkit for biomedicine applications.

Nanoantennas are nanoscale optical elements capable of tailoring light-matter interactions by controlling and manipulating light at unprecedented levels[1,2], allowing for exciting functionalities and cutting-edge applications across diverse fields such as photothermal therapy[3], optical communications[4], imaging[5], and sensing[6]. The light-matter interaction involving nanoantennas and molecules, specifically in terms of molecular refractive index and vibrational absorption, is particularly intriguing in the mid-infrared (IR) region[7]. This is because various biomolecular IR vibrational fingerprints can be observed through absorption spectra, which are associated with molecular constituents, chemical bonds, and intrinsic configuration[8]. The recent remarkable advancements in nanophotonic IR vibrational

spectroscopy show the potential to revolutionize disease diagnosis, treatment, and prevention[9], by enabling high-resolution biomolecular imaging[10], in-vitro molecular diagnosis[11], real-time biochemical monitoring[12], and more[13]. The utilization of artificial intelligence in sensory nanoantennas also offers valuable perspectives in biomedical science and measurable advancements in clinical diagnostics[14,15].

A major obstacle in infrared spectroscopy is the inability to identify molecules when infrared vibrational modes are overlapped[16]. Nanoantennas exhibit the capability of enhancing molecular vibrational modes through the surface-enhanced infrared absorption (SEIRA) effect[17–20]. However, it is challenging for SEIRA to disentangle overlapping vibrational modes. Great efforts have been dedicated to

[1]Department of Electrical and Computer Engineering, National University of Singapore, Singapore 117583, Singapore. [2]Center for Intelligent Sensors and MEMS (CISM), National University of Singapore, Singapore 117583, Singapore. [3]Key Laboratory of Optoelectronic Technology & Systems of Ministry of Education, International R&D Center of Micro-Nano Systems and New Materials Technology, Chongqing University, Chongqing 400044, P. R. China. [4]NUS Suzhou Research Institute (NUSRI), Suzhou, Jiangsu 215123, China. [5]NUS Graduate School—Integrative Sciences and Engineering Programme (ISEP), National University of Singapore, Singapore 119077, Singapore. ✉e-mail: mxjacj@cqu.edu.cn; elelc@nus.edu.sg

resolving the issue. Firstly, the IR fingerprint of a molecule typically comprises multiple vibrational modes. If the fingerprints only partially overlap, molecular identification could still be accomplished by examining additional features[21]. Secondly, the issue can be bypassed through the screening of target molecules from mixtures exhibiting overlapping vibrations, by employing either physical[22] or biochemical methodologies (e.g., specific binding)[23]. Furthermore, the utilization of sophisticated data processing techniques, such as derivative spectroscopy[24], deconvolution[25], or artificial intelligence algorithms[14], can contribute to discerning overlapping vibrational modes. Despite the practical efficacy of these methods, the conundrum pertaining to overlapping vibrational modes persists from a physical mechanism perspective. The molecular refractive index is intricately linked to electronic polarizability, which incorporates crucial features related to the composition and structural attributes of a material[26,27]. Consequently, the refractive index holds the potential to theoretically facilitate de-overlapping molecular fingerprints by offering supplementary features. Currently, the development of an effective methodology to achieve this goal is both desirable and challenging.

Surface phonon polaritons (SPhP) are a type of surface wave that is formed when polar optical phonons interact with long-wavelength fields[28]. Previous investigations focusing on SPhP have substantiated its remarkable optical characteristics, such as strong electromagnetic field confinement[29], slow group velocity[30], long lifetime[31], hyperbolic dispersion[32], and low loss[33]. Particularly, the low loss of SPhPs offers the advantage of high quality factors, surpassing those of their plasmonic counterparts[34]. The potential for sensitive molecular vibrational spectroscopy has been demonstrated through the use of boron nitride nanoantennas, achieving femtomolar sensitivity[35]. When SPhP is integrated with localized surface plasmon polariton (LSPP), the light-matter coupling strength is ultrastrong, comparable in magnitude with the vibrational frequency, referred to as vibrational strong coupling (VSC)[36]. VSC is regarded as a promising approach to address sensitivity issues in infrared vibrational spectroscopy, enabling progress toward ultrasensitive sensing applications[37]. However, to the best of our knowledge, SPhPs have not been exploited to address the aforementioned issue of overlapping vibrational modes.

Here, we show how to address the issue of overlapping vibrational modes by developing a stacked IR nanoantenna to excite coupling modes of localized surface plasmon and surface phonon polaritons (SP-PhP platform). Specifically, we discover that the SPhP vibration within the Reststrahlen band is sensitive to the molecular refractive index (RI), termed RI-dependent SPhP vibrations. Then, we experimentally examine the response of RI-dependent SPhP vibrations to the refractive index. Based on these findings, we showcase how the RI-dependent SPhP vibration, when combined with deep neural network algorithms, dynamically disentangles strongly overlapping molecular vibration in biological reactions, which is challenging for conventional IR spectroscopy techniques.

## Results
### Methodology
The concept of our SP-PhP platform for the dynamic identification of overlapping molecular vibrations in biological reactions is illustrated in Fig. 1a. The SP-PhP platform consists of stacked trapezoidal metal (Au) antennas and phonon (silicon oxide, $SiO_2$) antennas, all stacked on top of one another and situated on a $BaF_2$ substrate. The rationale behind the design is that both SPP and SPhP modes are confined strictly to the material's surface[36,38], and the stacking design facilitates a more effective spatial overlap of these two modes, thereby enhancing the mode coupling efficiency. The two polariton modes (LSPP and SPhP) are coupled to form polariton hybridization. Additionally, the two metal antennas of our SP-PhP platform form a parallel plate configuration separated by a dielectric medium. Upon being stimulated by infrared radiation, two plasmonic resonant modes become excited: a lower-

frequency bonding mode and a higher-frequency antibonding mode[39]. The bonding mode presents a displacement current with identical flow directions in the two parallel antennas, under the driving of a time-varying electromagnetic field. Conversely, the antibonding mode displays opposite current directions between the parallel antennas, forming a virtual current loop due to the unequal current intensities (Supplementary Fig. 1). We demonstrate that when the resonant wavelength of the two resonant modes are in proximity to each other, broadband resonance is obtained (1.5 times wider than nanorod, Supplementary Fig. 1b). Broadband resonances are desirable for infrared vibrational spectroscopy as they allow a broader range of molecular vibrations to be covered and enhanced through polaron-molecule coupling[40]. Additionally, akin to conventional antennas[41], the stacked antennas retain geometry-dependent characteristics (Supplementary Fig. 2). The splitting of the two modes can be controlled by engineering the thickness of metal antennas. When the thicknesses of mental antennas are equal (Supplementary Fig. 3b), an increase in the thickness of the spacer enlarges the splitting between the two modes. Conversely, when the thicknesses of mental antennas are not equal (Supplementary Fig. 3c), the light incident from the thicker antenna to the thinner antenna is easy to excite the mode splitting. Additionally, there is no significant difference observed in the polarization response and the angle of incidence range between the stacked antennas and conventional nanorods (Supplementary Figs. 4, 5).

The glucose enzymatic reaction (GER), is chosen as a proof-of-concept for investigating the dynamic identification of overlapping molecular vibrations. The GER is a process in which the glucose oxidase (GOD) catalyzes the conversion of glucose (**1**) into gluconic acid (**2**) and hydrogen peroxide ($H_2O_2$, **3**) (upper panel of Fig. 1b)[42]. In the field of biomedicine, all reactants involved in GER have demonstrated utility in cancer therapeutic strategies (see Methods for details)[43]. Given that molecular vibrations can furnish crucial conformational information about molecules[44], the dynamic analysis of these vibration-related bond-breaking-bond-making events along the reaction path is intriguing and meaningful for investigating such strategies. Moreover, there are overlapping modes between the O-H vibration (corresponding to $H_2O_2$) and the stretching vibration of amide groups (corresponding to GOD) (lower panel of Fig. 1b). Therefore, GER is the ideal candidate for our demonstration. Deep learning algorithms also serve as a powerful tool to aid in the demonstration. The GOD is immobilized on the surface of the SP-PhP platform through the chemical crosslinking of graphene oxide (GO), confining GER within the near-field region of the SP-PhP platform.

Next, we theoretically investigate the response of our SP-PhP platform to molecules. As mentioned earlier, our stacked SP-PhP platform possesses two polariton modes: LSPP and SPhP. The response of LSPP and SPhP to molecules varies significantly. The LSPP can enhance the vibrational IR absorption of molecules through the well-known SEIRA effect, which is related to the imaginary part of the complex refractive index[45]. Regarding SPhP, we discover that the SPhP in the polariton hybridization of our platform is sensitive to the real part of the complex refractive index (referred to as refractive index ($n$) hereafter). Notably, sensitivity to refractive index does not imply the simultaneous measurement of the real and imaginary parts of the composite refractive index in the same wavelength range. To validate our predictions, it is essential to establish the relationship between SPhP and $n$. The SPhP is excited in the "Reststrahlen" band, situated between the longitudinal (LO) and transverse optic (TO) phonon frequencies of polar dielectric crystals ($SiO_2$ in this work)[28]. We denote the vibrational variations of SPhP within the Reststrahlen band as $\Delta l$. Therefore, the objective becomes to establish the relationship between $\Delta l$ and $\Delta n$. To examine the $\Delta l$-$\Delta n$ relationship, we solve the wave equation assuming a simple harmonic solution for the wave quantity, and compute the dispersion curve for the polariton hybridization of our SP-PhP platform. The dispersion relation of surface plasmon polariton can be obtained by solving the Maxwell equations

with appropriate boundary conditions and is expressed as ref. [46]:

$$k_{spp} = \frac{\omega}{c} \sqrt{\frac{\varepsilon_1 \varepsilon_2}{\varepsilon_1 + \varepsilon_2}} \qquad (1)$$

where $\varepsilon_1$ ($\varepsilon_1 = \varepsilon_1' + i\varepsilon_1''$) and $\varepsilon_2$ ($\varepsilon_2 = \varepsilon_2' + i\varepsilon_2''$) are metal and surrounding media's frequency-dependent permittivity, respectively, and $c$ is the speed of light. When the plasmon is coupled with molecules ($\varepsilon_m = \varepsilon_m' + i\varepsilon_m''$), the effect permittivity of surrounding media is substituted by the molecule $\varepsilon_m$. For the case of $\varepsilon_m' > \varepsilon_2'$ ($\Delta n > 0$), a

frequency shift (redshift, $\omega_A \to \omega_B$) is observed in the dispersion curve (upper panel of Fig. 1c). Then, the dispersion relation for LSPP-SPhP hybridization can be derived via a coupled-harmonic-oscillator model, assuming a simple harmonic solution for the wave quantity (Supplementary Note 2):

$$\omega_{\pm} = \frac{\omega_{LSPP} + \omega_{SPhP}}{2} - i\frac{\gamma_{LSPP} + \gamma_{SPhP}}{4}$$
$$\pm \frac{1}{2} \sqrt{4g^2 + \left[(\omega_{LSPP} - \omega_{SPhP}) - i\frac{\gamma_{LSPP} - \gamma_{SPhP}}{2}\right]^2} \qquad (2)$$

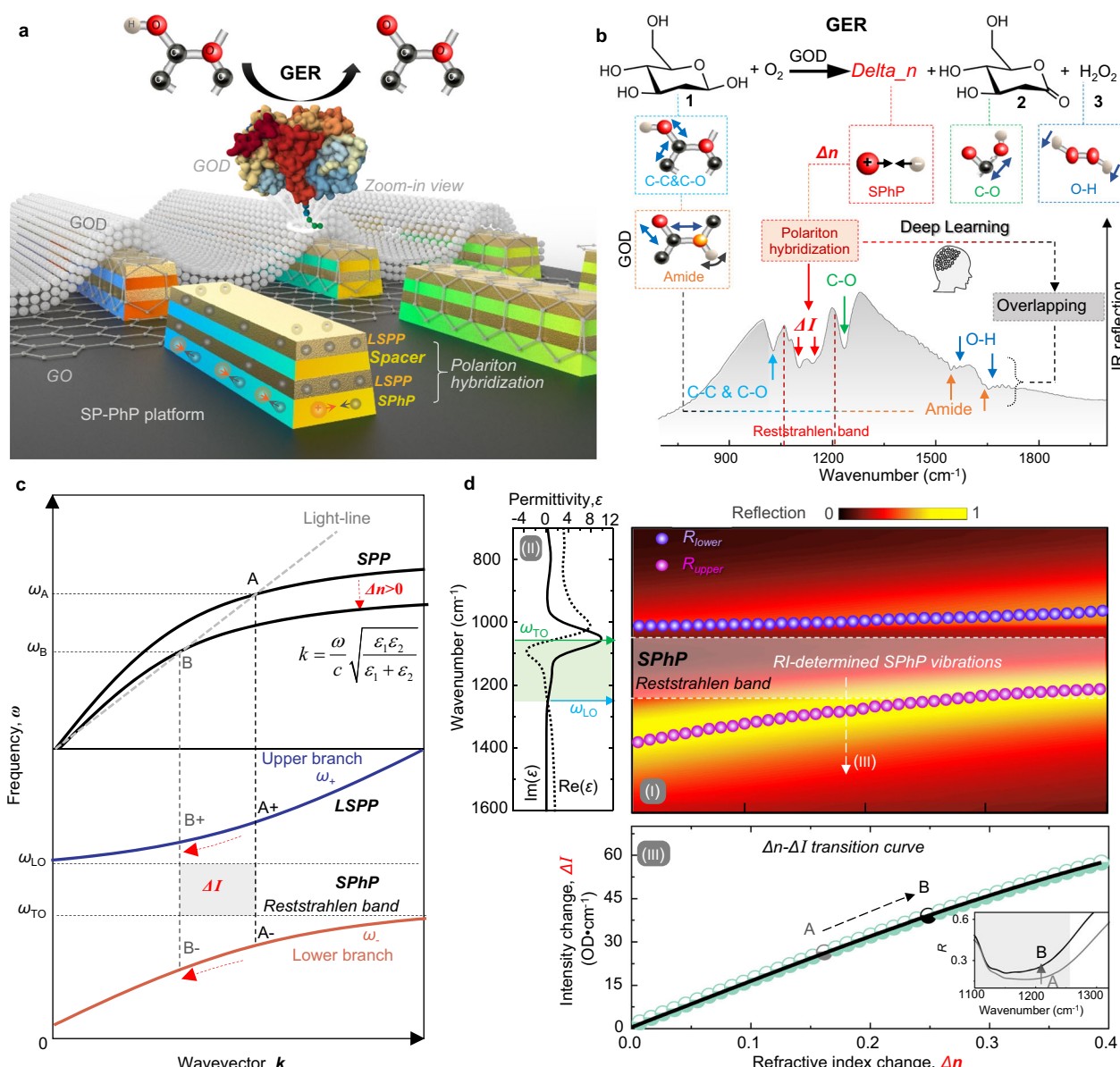

**Fig. 1 | Concept of RI-determined SPhP vibration for the identification of overlapping molecular vibrations. a** Schematic view of stacked IR nanoantennas (SP-PhP platform) for the identification of overlapping molecular vibrations in glucose enzymatic reaction (GER). RI-determined SPhP vibrations are excited. Glucose oxidase (GOD) is immobilized on the device surface through the chemical crosslinking of graphene oxide (GO). SPhP surface phonon polaritons, LSPP localized surface plasmon polariton. **b** Vibrational assignments of the GER reaction path. The O-H vibration and the amide vibration are overlapped. RI-determined SPhP vibrations, combined with deep learning algorithms, are used to identify the overlapping O-H vibrations and the amide vibrations. **c** Dispersion curves of SPPs (upper panel) and an LSPP-SPhP coupling system (lower panel). It describes how a

refractive index change ($\Delta n$) causes a change in SPhP vibrational intensity ($\Delta I$). The light-line represents the light that propagates along the surface. **d** Simulation results of the SP-PhP platform for molecules with varying $\Delta n$, investigating the relationship between $\Delta n$ and $\Delta I$. (I) The simulated reflection mapping. $R_{upper}$, $R_{lower}$: upper/lower polaritonic resonance. (II) The permittivity of SiO₂. (III) The $\Delta n$-$\Delta I$ transition curve obtained by calculating the intensity integral of the SPhP vibration within longitudinal ($\omega_{LO}$) and transverse optic ($\omega_{TO}$) phonon frequencies. The 3D model of GOD was imported from RSCB Protein Data Bank, DOI: 10.2210/pdb1CF3/pdb. The data was deposited by Hecht, H.J. and Kalisz, H. in 1999, under the title "GLUCOSE OXIDASE FROM APERGILLUS NIGER". 

where the variables include dressed frequencies ($\omega_{\pm}$), bare frequencies ($\omega_{LSPP}$ and $\omega_{SPhP}$), damping coefficients ($\gamma_{LSPP}$ and $\gamma_{SPhP}$), and coupling frequency splitting ($g$). For the molecular coupling ($\Delta n > 0$), its frequency response ($\omega_A \to \omega_B$) is analogous to that of surface plasmonic polariton in Eq. (1), because surface plasmonic polariton also exists in LSPP-SPhP hybridization. The distinction lies in the clear splitting observed in the dispersion curve of the LSPP-SPhP hybrid, situated in the Reststrahlen band of SPhP, as shown in the dispersion curve (lower panel of Fig. 1c, see Supplementary Fig. 6 for detailed derivation). Different positions on the split imply varying plasmonic resonance frequencies. As the SPhP Reststrahlen band remains fixed, altering the plasmonic resonance frequency $\Delta\omega$ yields distinct coupling strengths, consequently influencing the SPhP vibration intensity $\Delta I$. For instance, when the plasmonic resonance frequency changes from $\omega_A$ to $\omega_B$, the split moves from A + A- to B + B-. The change in the SPhP vibration intensity can be calculated by $\Delta I_{A,B} = \int_{\omega_{TO}}^{\omega_{LO}} D_{A,B}(\omega)d\omega$, where the variables $\omega_{LO}$ and $\omega_{TO}$ denote the TO and LO phonon frequencies of the $SiO_2$, respectively, and $D_{A,B}(\omega)$ are the corresponding differential spectrum of SPhP. Therefore, the relationship between $\Delta D$ and $\Delta n$ is the key to establishing the relationship between $\Delta I$ and $\Delta n$. Based on the understanding of the physical mechanism, $\Delta n$ leads to the redshift of the plasmonic resonance frequency, while the SPhP vibration remains fixed. Consequently, this redshift causes the detuning between the SPhP vibration and plasmonic resonance, resulting in a change in the SPhP spectral signal $\Delta D$ (Supplementary Note 3). Similar conclusions about the detuning of molecular vibrations and plasmonic resonances by frequency shifts can be found in the literature[17,47]. The relationship between $\Delta D$ and $\Delta n$ can be calculated and visualized by a finite-difference time-domain (FDTD) simulation. As observed in the simulated reflection mapping of refractive index versus wavenumber (Fig. 1d(I)), the reflection in the SPhP Reststrahlen band (corresponding to $\Delta D$) varies with $\Delta n$. By integrating $\Delta D$, we obtain the relationship between $\Delta I$ and $\Delta n$ as $\Delta I = 210.3 - 210.6 \cdot \exp(-\Delta n/1.22)$ with fitting degree $R^2 = 0.999$ (Fig. 1d(III)). The established $\Delta I$-$\Delta n$ relationship indicates that our SP-PhP platform can leverage the vibrational variations $\Delta I$ of SPhP within the Reststrahlen band (RI-dependent SPhP vibrations) to provide the molecular refractive index feature $\Delta n$. Therefore, the RI-dependent SPhP vibration is promising to address the issue of overlapping vibrational modes, as we demonstrated in the subsequent section.

**Experimental characterization of RI-dependent SPhP vibrations**
Following the theoretical prediction of the RI-dependent SPhP vibrations, we proceeded with a two-part experimental characterization. The first part is experimental observations of the RI-dependent SPhP vibrations to verify our prediction. The second part aims to explore the sensing performance of the RI-dependent SPhP vibrations. Regarding the experimental observations ($\Delta n \to \Delta I$), we realize them through two steps. The first step is to confirm that the SPhP vibration variation $\Delta I$ is caused by detuning $\delta$ of plasmonic resonance and SPhP vibration ($\delta \to \Delta I$), that is, the underlying physical mechanism of the RI-dependent SPhP vibrations is $\delta$. The second step involves confirming that the $\delta$ is determined by the refractive index change $\Delta n$ ($\Delta n \to \delta$). As shown in Fig. 2a, LSPP is coupled to SPhP through the medium of the near field. The vibration frequency of SPhP is fixed[48], while the resonance frequency of LSPP can be tuned by engineering the dimensions of metal antennas[49,50]. Therefore, by engineering the dimensions to control the resonance frequency of LSPP, various $\delta$ can be achieved as required. According to this consideration, in the first step, we designed antennas with resonances ranging from 900 to 1700 cm$^{-1}$ (orange dotted line in Fig. 2b), covering the reststrahlen band of the SPhP vibration from $\omega_{TO}$ to $\omega_{LO}$ (black dotted lines in Fig. 2b). In the crossing region, we observed an avoided crossing phenomenon resulting in resonance splitting into two branches, known as mode splitting (blue and brown dots in Fig. 2b)[46,51]. The splitting degree reflects the

coupling strength of LSPP-SPhP hybridization[46,52], and it shows a thickness-dependent transition (Supplementary Fig. 13). Our calculations (Supplementary Table 2) indicate that in the 100 nm thick SPhP antenna, the normalized coupling strength $\eta$ of polariton hybridization exceeds 0.1, reaching 0.138, which demonstrates that the system operates within the ultrastrong coupling regime[51]. This is appealing for sensing as it represents ultrastrong light-matter interactions that can withstand the high absorption loss of the biological environment, enabling robust molecule detection. Furthermore, each set of splitting frequencies ($\omega_+$, $\omega_-$) is distinct, implying that each split possesses a different detuning. Taking three splits (labeled A, B, C) as an example, the corresponding measured spectra are shown in Fig. 2c. Evidently, a variation $\Delta I_{BC}$ in the RI-dependent SPhP vibrations is observed when the detuning is changed from $\delta_B$ to $\delta_C$. Therefore, we conclude that $\Delta I$ is caused by $\Delta\delta$.

In the second step, we eliminate geometry-induced $\delta$ by fixing the antenna dimension and generate $\Delta n$ by varying the thickness of bovine serum albumin (BSA), to investigate its contribution to $\delta$ ($\Delta n \to \delta$). Figure 2d, e shows the SEM and AFM image of the SP-PhP platform with fixed average antenna length $l = 3$ μm. The outline of the pattern was well-defined, demonstrating the effectiveness of the fabrication process. Figure 2f shows the cross-sectional SEM view and EDX mappings of three key elements. Each layer was tightly bonded and quite distinct from each other. Then, the SP-PhP platform was spin-coated with different thicknesses of BSA molecules (Supplementary Fig. 14). The $\Delta n$-induced mode splitting was observed in the measured spectra (Fig. 2g), where splitting frequencies ($\omega_+$, $\omega_-$, dotted line) is distinct, implying different $\delta$. After baseline calibration, the RI-determined SPhP vibrations were experimentally observed (Fig. 2h(I)), while the stretching vibrations of amide I and II of BSA were also detected due to the SEIRA effect. By integrating the RI-determined SPhP vibrations (Fig. 2h(II)), we obtain the relationship as $\Delta I = \exp(2.77 + 3.72 \cdot t_{BSA} - 2.1 \cdot t_{BSA}^2)$, wherein we use thickness $t_{BSA}$ instead of the $\Delta n$, as $\Delta n$ is a complex frequency-dependent function, and the thickness is easy to control and also related to the effective refractive index[53]. Next, we analyze the sensitivity optimization of the system using the radiating oscillator model (Supplementary Note 2). The 2D sensitivity mapping calculation demonstrates that the maximum sensitivity value is in the weak coupling regime (Fig. 2i). The robustness of strong coupling systems and their reduced sensitivity to $\Delta n$ could be the reason behind this phenomenon.

Next, we experimentally examine the sensing performance of the RI-dependent SPhP vibrations. The chip (the SP-PhP platform or nanorods) was integrated with the microfluidic system in an internal reflection manner (Fig. 3a(I, II))[14,15], enabling molecular detection in aqueous environments. Upon injecting 300 mM glucose solutions (Fig. 3a(III)), the nanorod responds according to the refractive index-induced frequency shift mechanism. We designed and fabricated two antennas with different bandwidths. A 5.2 cm$^{-1}$ shift was observed for the narrowband nanorod (Nanorod 2 in Fig. 3b(I)), while no shift for the broadband nanorod (Nanorod 1 in Fig. 3b(I)) due to its low figure of merit (FOM). With the FTIR spectrometer's resolution of 4 cm$^{-1}$, the limit of detection (LoD) of nanorods is 300 mM. For the same concentration of analyte (Fig. 3b(II)), our SP-PhP platform exhibited a considerable intensity change ($\Delta I = 4802$ mOD·cm$^{-1}$), which is above our platform's detection limit. To investigate the detection limit, we performed a series of experiments with glucose concentrations varying from 0 mM to 80 mM. The extracted differential reflection spectra are shown in Fig. 3c(I). Apparently, the intensity of differential reflection rises with the increase in glucose concentration. By integrating the differential reflection, we obtain the sensing curve of total molecular signal versus concentration (Fig. 3c(II)). Taking the total noise into account (Supplementary Fig. 15), LoD for glucose sensing on the SP-PhP platform is 5 mM, surpassing that of the nanorod by 60 times. In addition to sensitive sensing, another discovery in the RI-dependent

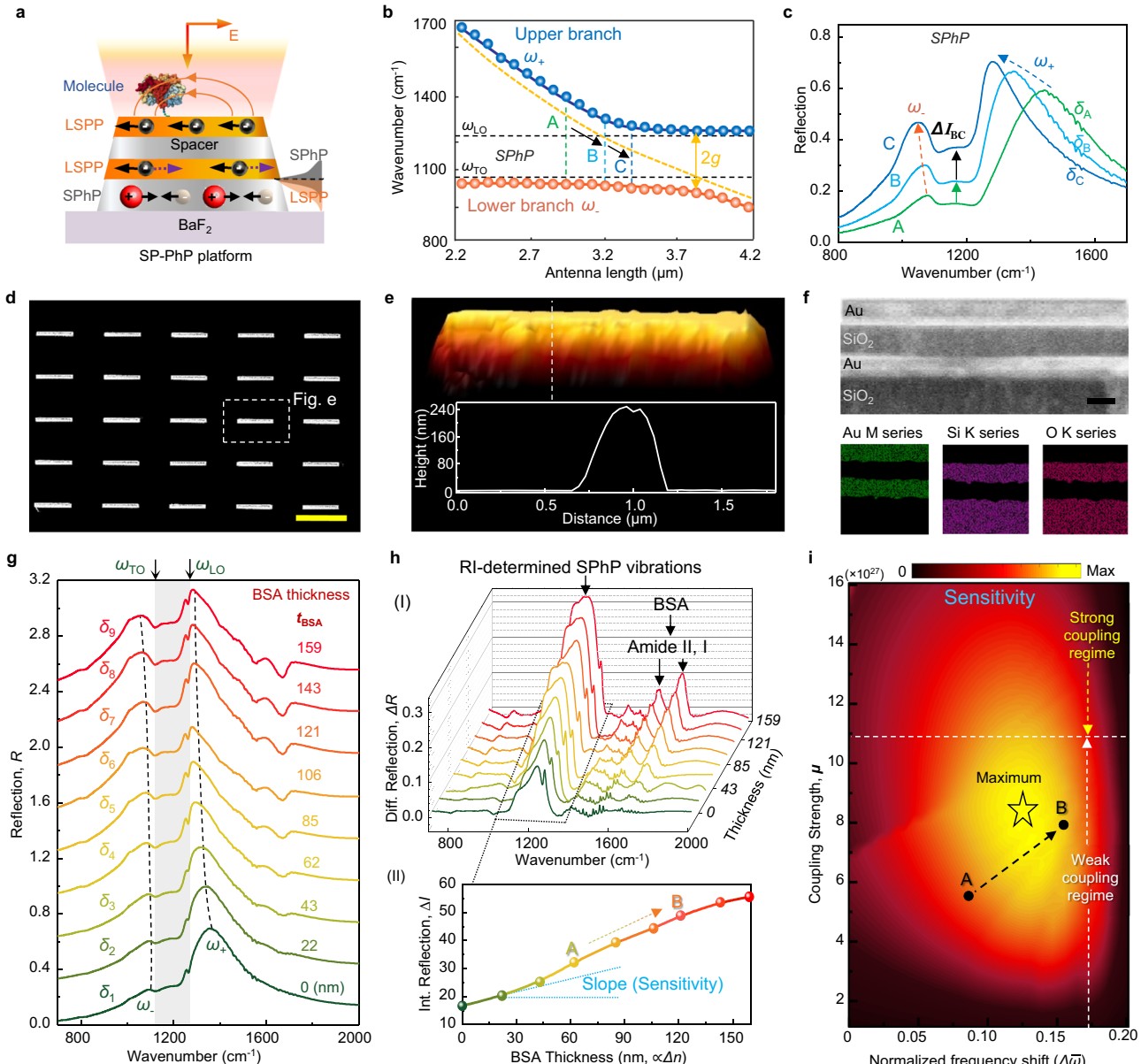

**Fig. 2 | Experimental observation of RI-dependent SPhP vibrations. a** Schematic of LSPP-SPhP coupling in the SP-PhP platform. **b** The dispersion relation of the SP-PhP platform plotted by experimentally measured data. Various detuning ($\delta$) of plasmonic resonance and SPhP vibration is obtained by engineering the antenna length. The letters A, B, and C label three specific $\delta$. **c** Measured spectra of the SP-PhP platform corresponding to A, B, and C. The change from $\delta_B$ to $\delta_C$ causes a variation in the RI-dependent SPhP vibrations ($\Delta I$). **d, e** SEM and AFM images of the SP-PhP platform with average antenna length $l = 3\,\mu m$. Scale bar, 4 $\mu m$. **f** Cross-sectional view (upper panel) of the SP-PhP platform using SEM. Lower panel:

corresponding EDX mappings of three key elements for observing Au and SiO$_2$ layers. **g** Measured spectra of the SP-PhP platform when coupled to BSA molecules of varying thicknesses. **h** Corresponding baseline-corrected molecular signals of the SP-PhP platform. (I) Two types of signals are observed, one being the RI-determined SPhP response and the other being the vibration mode of the bovine serum albumin (BSA) molecule (amide I, II). (II) The relationship between $\Delta I$ and BSA thickness. **i** Calculated two-dimensional mapping of sensitivity as a function of coupling strength and normalized frequency shift.

SPhP vibrations is joint baseline fitting. Baseline correction is crucial in spectral data analysis to ensure the accurate extraction of spectral signals[54]. Different baseline fittings can distort signals, leading to (1) inaccurate refractive index measurements due to frequency drift discrepancy (Fig. 3d-I), and (2) failure of fingerprint extraction because of molecular vibration intensity discrepancy (Fig. 3d-II). These discrepancies arise because only the signal's shape and physical characteristics are available for baseline fitting. Interestingly, the RI-dependent SPhP vibrations can provide information on molecular refractive index. As depicted in Fig. 3d-III, when the isopropyl alcohol (IPA) molecules are introduced to our SP-PhP platform, two responses

are observed: a frequency shift $\Delta f$ and a corresponding $\Delta I$. The $\Delta I$ can be utilized to accurately correct $\Delta f$ for a precise fitting curve, which is challenging in conventional plasmonic devices.

**DL-assisted vibrational de-overlapping in bioreaction profiling**
Next, we demonstrate how these unique properties of RI-dependent SPhP vibrations can be utilized to address technical challenges of overlapping vibrational identification in infrared biospectroscopy. The glucose enzymatic reaction (GER) is chosen as the target. Initially, we functionalized the SP-PhP platform by immobilizing GOD on the device surface through a chemical cross-linking method (Fig. 4a).

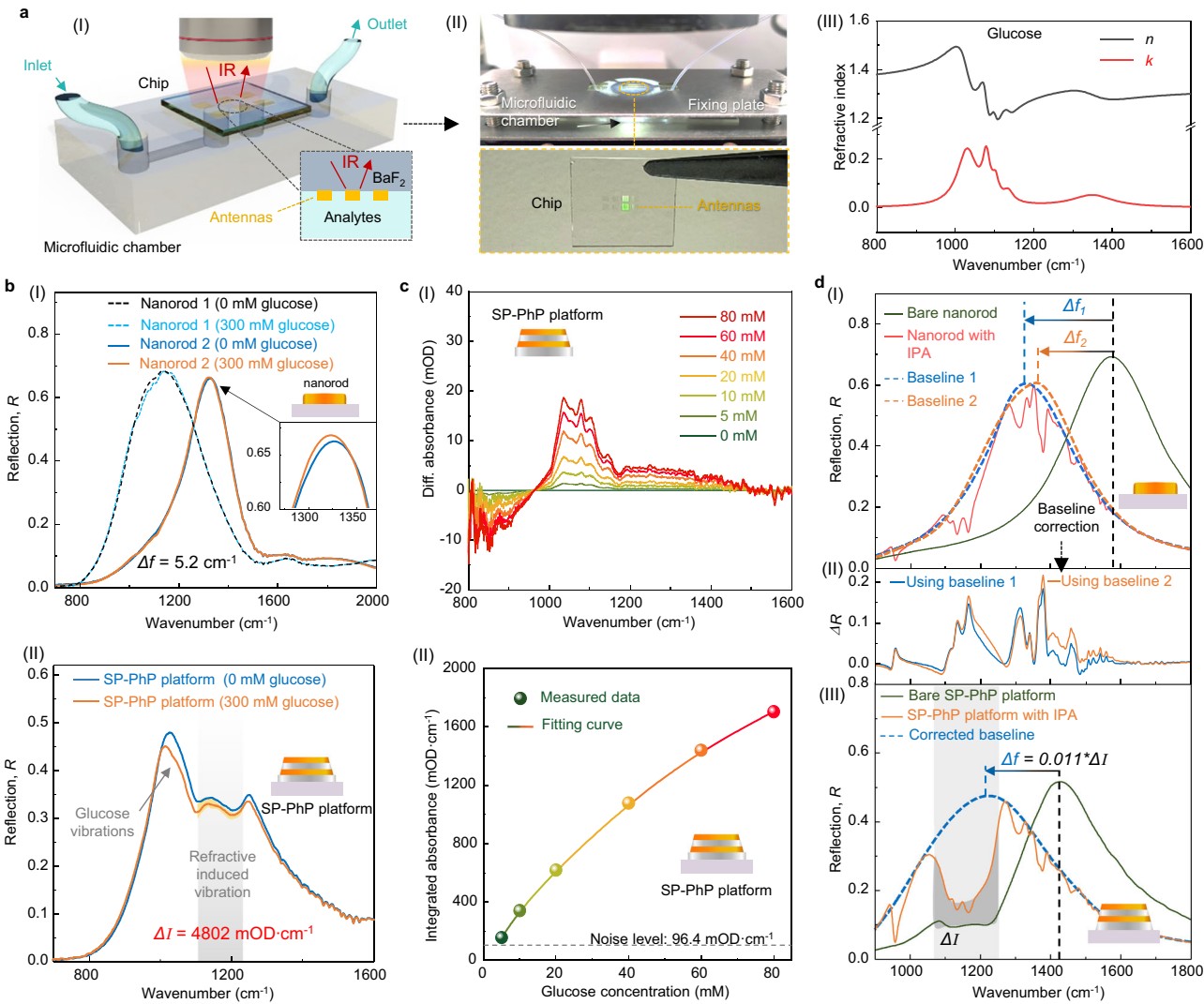

**Fig. 3 | Biosensing and joint baseline fitting using RI-dependent SPhP vibrations. a** Experimental configuration. (I) Schematic of the chip integrated with PDMS microfluidic chamber for analyte detection. (II) Photo of the experimental setup. (III) Complex refractive index plotting of the analyte (glucose), showing details of refractive index ($n$) and extinction coefficient ($k$). **b** Comparison of glucose detection on (I) nanorod platform based on the common sensing principle of refractive index-induced frequency shift ($\Delta f$) and (II) SP-PhP platform based on RI-determined SPhP vibrations. **c** Sensitive glucose detection using the SP-PhP platform when glucose concentrations vary from 0 mM to 80 mM. (I) Reflection spectra and (II) baseline-corrected differential absorbance signals are shown.
**d** Comparison of baseline fitting methods. (I) Common baseline fitting for nanorods. The baseline fitting is affected by model parameters, and different baselines are obtained due to the overfitting of the spectrum. (II) Corresponding signal discrepancy caused by baselines. (III) Joint baseline fitting of the SP-PhP platform. The RI-determined SPhP vibration signal joint with an asymmetric least square algorithm (Supplementary Note 4) is used for the accurate baseline fitting of a measured spectrum, eliminating baseline discrepancy.

Successful immobilization is confirmed by the observation of amide vibrations of GOD in the measured spectra (Fig. 4b). It is then integrated with a microfluidic system (Fig. 4c), and 110 mM glucose solutions were injected to initiate the GER experiment. Subsequently, a series of time-varying reflectance spectra at 1-min intervals were collected by our SP-PhP platform (Fig. 4d). Notably, the environmental disturbances can be eliminated by measuring the background spectrum before each spectrum collection (Supplementary Fig. 16). Utilizing joint baseline correction in conjunction with an asymmetric least squares (ALS) algorithm (Supplementary Note 4), we extracted the molecular vibration signals and RI-dependent SPhP vibration signals (Fig. 4e). As observed, the O-H vibration (corresponding to $H_2O_2$) and the stretching vibration of amide groups (corresponding to GOD) strongly overlap, making it challenging to identify them based on their individual features. To achieve higher resolution monitoring, we set the time interval for collecting spectra to 20 s. Figure 4f shows the

obtained differential spectral signal. However, it is apparent that the issue of overlapping vibrations has not been improved.

To address the issue of overlapping vibrations, we develop a neural network (DNN) model[55] to process the molecular vibration signals and RI-dependent SPhP vibration signals. The DNN model consists of input, hidden, and output layers (Methods). To train the DNN model, we performed single-analyte experiments measuring only one analyte at a time. The obtained spectra build a signature library of RI-determined SPhP vibrations for the GER reactants (Fig. 5a). As observed, the molecular vibration signals and RI-dependent SPhP vibration signals corresponding to each analyte exhibit distinct features. Then, the data obtained from single-analyte experiments and multiple-analyte experiments (Supplementary Note 5) were utilized to train the DNN models. The training and validation dataset comprises 20,156,640 spectrotemporal data points. The accuracy of the trained DNN model reaches 92% (Supplementary Fig. 9). Subsequently, the

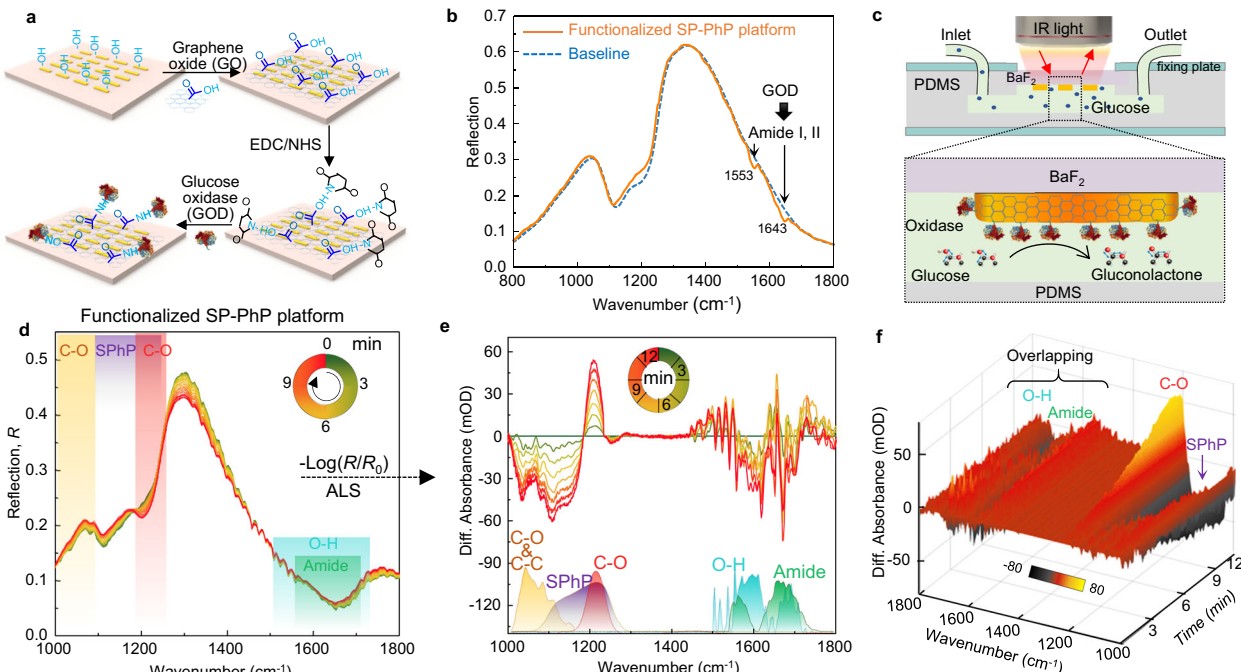

**Fig. 4 | Dynamic profiling of bioreactions with overlapping vibrations.**
**a** Schematic illustration of the GOD functionalization of the SP-PhP platform by chemical crosslinking of GO. EDC 1-ethyl-3-(3-dimethylaminopropyl) carbodiimide, NHS N-hydroxysuccinimide. **b** The spectrum of the functionalized SP-PhP platform, indicating the successful bonding of GOD. **c** Schematic of the microfluidic experimental setup integrated with the functionalized chip for GER profiling. **d** Collected real-time reflection spectra of the functionalized platform when

starting GER by injecting 110 mM glucose solutions. Time interval: 1 min. **e** 2D plots of differential absorbance spectra calculated as $-\log_{10}(R/R_0)$, where $R$ and $R_0$ are the measured and baseline reflection, respectively. Lower curves are scaled absorbance spectra of the reactant vibration modes in GER, where some modes are overlapping. **f** 3D plots of differential absorbance spectra versus wavenumber and time. Time interval: 20 s.

measured real-time spectral data obtained during the GER experiment (Fig. 4f) were fed into the trained DNN model (Fig. 5b) to output prediction (weights) for all reactants in the reaction. In the predicted result (Fig. 5c(II)), we can observe that all bond-breaking-bond-making events in the GER experiment (Fig. 5c(I)) were fully identified. Particularly, we successfully disentangled the strongly overlapping vibration modes between the O-H vibration and the stretching vibration of amide groups (Line ii and Line iv in Fig. 5c(II)). As the reaction progresses, glucose (**1**) is consumed and gluconolactone (**2**) and $H_2O_2$ (**3**) are continuously accumulated. Consequently, the vibration signals of C-O and C-C bonds (glucose, Line i in Fig. 5c(II)) decrease, while the vibration signals of C-O bonds (gluconolactone, Line iii in Fig. 5c(II)) and O-H bonds ($H_2O_2$, Line iv in Fig. 5c(II)) climbed. As GOD acts as a catalyst in this context, the vibration signal of its amide bond remains essentially unchanged (Line ii in Fig. 5c(II)). The predicted results align with the enzyme kinetics findings reported in the previous literature[56].

For comparison, we employed a conventional nanorod-based platform to monitor the GER reactants. The nanorod-based platform lacks the capability to acquire the RI-dependent SPhP vibration signals. Subsequently, the nanorod-based platform is employed for single-analyte experiments (Supplementary Fig. 10), enabling the acquisition of IR vibrational features of the analytes. The configuration of the DNN model for the nanorod-based platform is identical to that of our SP-PhP platform. Then, the acquired data were employed to train the DNN model. Upon training and validation, the real-time spectral data newly obtained by the nanorod-based platform (Fig. 5d) were used to predict the output weights. In the predicted result (Fig. 5e), as the reaction progresses, the vibration signals of C-O and C-C bonds (glucose, Line i in Fig. 5e) decrease, while the vibration signals of C-O bonds (gluconolactone, Line iii in Fig. 5e), O-H bonds ($H_2O_2$, Line iv in Fig. 5e), and the amide bonds (Line ii in Fig. 5e) climbed. Compared to the results of our SP-PhP platform in Fig. 5c, there was an abnormal increase in the

amide vibration intensity and only a slight climb in the H-O vibration intensity. The increase in the amide vibration intensity means that the amount of GOD has increased. Indeed, it is incorrect because GOD, being an enzyme, functions as a catalyst in the reaction. Therefore, the reason could be attributed to the DNN model's inability to entirely distinguish overlapping vibrations, leading to an erroneous assignment of the H-O vibrational mode to the amide vibrational signal. It can also account for the slight climb of the H-O vibration, which should ideally exhibit a rapid rise. In our SP-PhP platform, the predicted difference between the overlapping O-H bond and amide bond is 0.5 (the difference between Line ii and Line iv at time 12 min in Fig. 5c(II)), whereas in the nanorod platform, it is merely 0.1 (Fig. 5e). This indicates that the DNN model trained by the data of our SP-PhP platform possesses the capability to identify the overlapping vibrations, while that trained by the data of the common nanorod platform fails in such identification and erroneously assigns H-O vibration modes to amide vibration signals. The feature of the SP-PhP platform compared to the common nanorod platform is the presence of RI-dependent SPhP vibrations. Therefore, we conclude that RI-determined SPhP vibrations can decouple overlapping vibrational modes. In contrast to previously reported infrared antenna techniques for biospectroscopic applications, our platform represents an advancement in accurately resolving overlapping vibrational modes (Supplementary Table 3).

## Discussion

We have demonstrated the dynamic construction of RI-dependent vibrations using surface plasmon-phonon polaritons in the SP-PhP platform, enabling precise accurate identification of the strongly overlapping vibrational modes. Our contribution is twofold. The first is the construction and experimental observation of RI-dependent vibrations. This vibration is a type of SPhP oscillation occurring within the reststrahlen band of polar dielectric crystals, exhibiting a

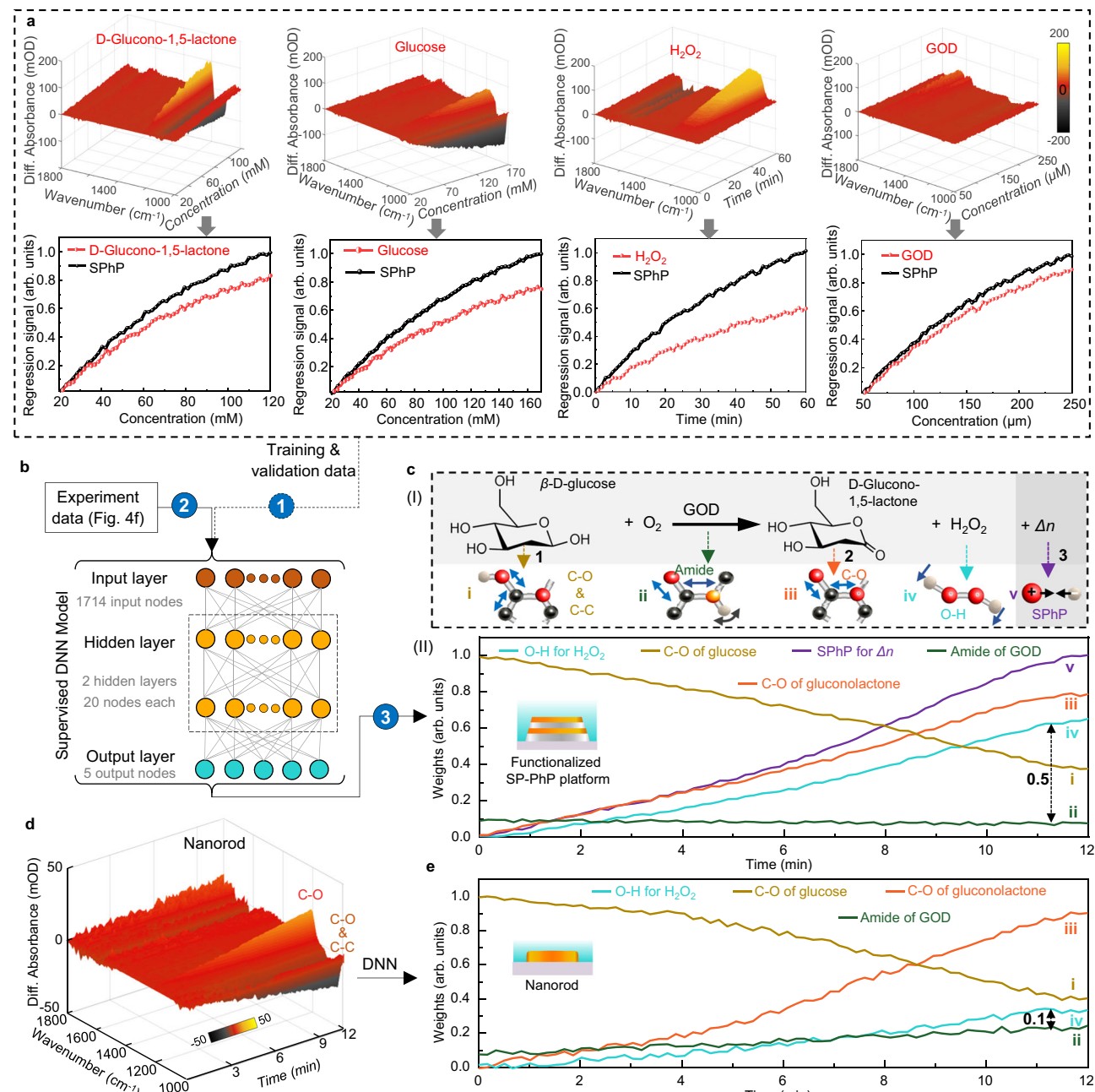

**Fig. 5 | DL-assisted analysis of bioreactions with overlapping vibrations. a** Real-time 3D plots of differential absorbance spectra versus wavenumber and concentration when using the SP-PhP platform to detect one analyte at a time (upper panel), including glucono-1,5-lactone (**2**), glucose (**1**), H₂O₂ (**3**), and GOD. Lower panel: Corresponding regression curve. **b** The DNN model with two hidden layers and 20 nodes for identification of overlapping vibrations in GER profiling. Step 1: training and validation data are used to train the model. Step 2: the experiment data is put into the trained model. Step 3: output the prediction for all vibrations monitored. **c** Predicted results of the DNN model based on the SP-PhP platform. The (I) is the monitored vibration in the GER reaction path, and the (II) is the predicted results. The difference between Line ii and Line iv at time 12 min is 0.5. **d** The real-time spectral data of a functionalized nanorod platform for GER monitoring. The DNN model training process is the same as that in (**b**). **e** The prediction of the DNN model based on the nanorod platform. The difference between Line ii and Line iv at time 12 min is 0.1.

notable sensitivity to the molecular refractive index. Our experimental demonstration elucidates the fundamental sensitivity mechanism of these vibrations to refractive index changes, which is governed by the refractive index-induced detuning between the SPhP vibration and plasmonic resonance. Besides, compared with conventional plasmonic antennas (nanorods), the RI-dependent vibrations show 1) potential in resolving molecular refractive index feature (Fig. 2h), 2) high-precision joint baseline correction (Fig. 3d), and 3) a 60-fold improvement in LoD, reaching physiological levels of 5 mM in the glucose detection (Fig. 3b, c). Notably, the real part and imaginary part of the complex

refractive index are linked by Kramers-Kronig relations. However, accurately converting the imaginary part into the real part using SEIRA signals is challenging in practice (Supplementary Note 6). Furthermore, the coupling between LSPP and SPhP has no special restrictions, its response to the refractive index is also general.

The second is the technological advancement of de-overlapping vibrational modes. We experimentally reveal that the RI-determined SPhP vibrations, combined with DNN algorithms, can decouple overlapping IR vibrational modes. In the demonstration of dynamic profiling glucose enzymatic reaction, we observe all bond-

breaking-bond-making events with a 92% identification accuracy, even for strongly overlapping vibration modes. This level of accuracy surpasses what can be achieved with common plasmonic platforms. Specifically, in our SP-PhP platform, the predicted difference between the overlapping vibrations - a metric reflecting its identification capability for overlapping modes—reaches 0.5, whereas for the common plasmonic platforms, it is only 0.1 (Fig. 5c, e). Our decoupling technique holds promise for biomedical applications such as molecular screening, protein conformation and dynamics resolving, and pharmaceutical analysis, owing to the presence of molecular massive overlapping vibrational modes in these applications.

Looking forward, our current on-computer data processing and decision-making can be developed for near-sensor or in-sensor computing to improve the accuracy and speed of detection[57]. Additionally, the implementation of miniaturized spectrometers to empower our technique towards point-of-care testing (POCT) would also be impactful[7,9,11]. Besides, the concept of surface plasmon-phonon polariton coupling is universal and could be implemented in other material systems, such as two-dimensional van der Waal's crystals (e.g., hexagonal boron nitride[48]) and polar semiconductors (e.g., gallium nitride)[28]. Finally, our de-overlapping concept could be extended to other reactions involving overlapping vibrations, like protein condensation reactions[58].

# Methods

## Numerical simulations

A commercial software package (FDTD Solutions, Lumerical Inc) based on the FDTD method was utilized for both spectral and near-field simulations. The incident radiation source was a plane wave light source with polarization oriented along the arm direction of the nanoantennas. The periodic boundary condition was employed to represent the periodicity of the nanoantennas in the x- and y-directions, while a perfectly matched layer was applied in the z-direction. We omitted the thin titanium (Ti) adhesion layer from the simulation and computed the spatial profile of the E-field intensity using a 3D frequency-domain power monitor. The refractive index of $BaF_2$ was set to 1.4, and the refractive index of Au was obtained from Palik et al. $SiO_2$ permittivity in Fig. 1 and the complex refractive index of glucose in Fig. 3 were calculated based on its measured IR absorption spectrum using the Drude model $\varepsilon(\omega) = \varepsilon_\infty + \sum_i^n \frac{S_i}{\omega_{0i}^2 - \omega^2 - j\omega\gamma_i}$, where $\gamma_i$ refers to the damping frequency, $S_i$ represents the oscillator strength, $\omega_{0i}$ denotes the oscillator resonance frequency, and $\varepsilon_\infty$ is the high-frequency constant term (See Supplementary Note 7 for details). In the simulation of calculating the $\Delta n$-$\Delta I$ converting curve, the thickness of analytes is set to 200 nm, and the $\Delta n$ changes from 0 to 0.4. The average length of the platform is defined as the length at half of the maximum height.

## Theory modeling

The dispersion curve in Fig. 1c and the 2D mapping of sensitivity as a function of coupling strength and normalized frequency shift in Fig. 2i are calculated using the radiating oscillator model. The equations of motion describing the coupled system are determined by the following relationship:

$$\begin{bmatrix} \omega_{LSPP} - \omega - \frac{i\gamma_{LSPP}\omega}{\omega_{LSPP}+\omega} & \frac{\kappa_{LSPP}}{\omega_{LSPP}+\omega} \\ \frac{\kappa_{LSPP}}{\omega_{LSPP}+\omega} & \omega_{SPhP} - \omega - \frac{i\gamma_{SPhP}\omega}{\omega_{SPhP}+\omega} \end{bmatrix} \begin{bmatrix} p(t) \\ q(t) \end{bmatrix} = H \cdot \begin{bmatrix} p(t) \\ q(t) \end{bmatrix} = \begin{bmatrix} 0 \\ 0 \end{bmatrix}$$

where $\gamma_{LSPP,SPhP}$, and $\omega_{LSPP,SPhP}$ are the damping factor and resonance frequency, respectively. The two resonators ($p(t)$ and $q(t)$) are coupled linearly via coupling strength $\kappa_{LSPP}$. To obtain the solution, we can solve the characteristic equation $|\mathbf{H} - \omega\mathbf{I}| = 0$, where $\mathbf{I}$ is the identity matrix. The dressed frequencies of the system can be expressed as

$$\omega_\pm = \frac{\omega_{LSPP} + \omega_{SPhP}}{2} - i\frac{\gamma_{LSPP} + \gamma_{SPhP}}{4}$$
$$\pm \frac{1}{2}\sqrt{4g^2 + \left[(\omega_{LSPP} - \omega_{SPhP}) - i\frac{\gamma_{LSPP} - \gamma_{SPhP}}{2}\right]^2}$$

where $g = \omega_+ - \omega_-$ is the frequency splitting. More derivation details are provided in Supplementary Note 2.

## Device nanofabrication

The fabrication process is shown in Supplementary Fig. 17. For the nanofabrication of the SP-PhP platform, the process commenced by cleaning the $BaF_2$ wafer in acetone through ultrasonic treatment for 10 min. After rinsing in isopropanol and drying with nitrogen, the wafer underwent oxygen plasma treatment for 5 min. Then, we spin-coated a 400 nm thick layer of PMMA e-beam lithography resist (950 PMMA A5) at 3000 rpm. After thermal bakes, a commercial electron-conducting polymer (Espacer 300Z from Showa Denko Singapore) was spin-coated at a speed of 2000 rpm to eliminate charge accumulation during e-beam exposure. The nanoantenna pattern was then exposed using e-beam lithography. After exposure, the sample was developed with deionized water, MIBK/IPA (1:3) mixture, and isopropanol. Then, $SiO_2/Ti/Au/Ti/SiO_2/$ Ti/Au films with desired thicknesses were sequentially deposited using e-beam evaporation, followed by immersion in acetone for 24 h to remove the unexposed resist and obtain the final nanoantenna pattern. The nanorod fabrication process is similar to that of the SP-PhP platform, with the exception that after exposure, only Ti and Au need to be deposited using e-beam evaporation.

## Optical measurement

The IR spectrum of our platform was measured using a Fourier transform infrared spectrometer coupled with an infrared microscope equipped with a liquid-nitrogen-cooled mercury cadmium telluride detector. The measurement area of the antenna array was controlled by knife-edge apertures and set to $100 \times 100 \, \mu m^2$. A resolution of $4 \, cm^{-1}$ and 20 scans per measurement was used. In the real-time experiment, 5 scans per measurement were used. The gold mirror was used to collect background data in reflection mode. To protect the optical path, nitrogen gas was continuously introduced into the microscope using designed accessories. The microfluidic channel utilized in the sensing experiment is fabricated using a 3D-printed mold and is affixed to a microscope slide. The chip is then integrated with the microfluidic system by flipping it over and aligning the nanoantenna with the channel. The infrared light passes through the $BaF_2$ substrate, interacts with the nanoantenna, and subsequently reflects back to the photodetector.

## Data processing and deep learning

The differential absorbance spectra were calculated by subtracting the measured reflectance spectra from the baseline fitted by our joint baseline fitting method. ALS algorithms were also used. The integrated absorbance signal was calculated by integrating the extracted vibrational bands. The intensity change of RI-determined SPhP vibration was calculated by integrating the spectral intensity change over the SPhP reststrahlen band. The figure of merit (FOM) is defined as Sensitivity/FWHM, where FWHM is full-width at half maximum. For the GER experiment, the reflectance spectrum at the beginning of the reaction was set as the background spectrum to calculate differential absorbance spectra. Vibrational signals of biomolecules were extracted by normalizing the reflectance spectrum of the chip to that before the reaction started using $-\log_{10}(R/R_0)$. The normalized frequency shift $\Delta\varpi$ was calculated as $(\varpi - \varpi_0)/\varpi_0$, where $\varpi$ and $\varpi_0$ are shifted and initial frequencies, respectively. The DNN model consists of input, hidden,

and output layers. The input and output layers comprise 1714 and 5 nodes, respectively, matching the spectral wavenumber points and reactants' vibrational numbers. The model comprises two hidden layers, each consisting of 20 nodes, dedicated to extracting high-level features from the input data. This design facilitates the learning of intricate relationships, leading to accurate predictions. The steps for using a DNN model to solve a regression problem involved preparing the data, designing the model architecture, training the model, validating its performance, and using it for prediction. In this work, the DNN model was developed using Keras, a Python framework, and consists of fully connected layers with ReLu as the activation function. The mean square error was used as the loss function and Adam is employed as the optimizer in the final model. The detailed processes are provided in Supplementary Note 5.

### Glucose enzymatic reaction experiment

All reactants in GER have potential applications in cancer diagnosis and treatment. Glucose consumption can be used as an alternative strategy for cancer starvation therapy. Oxygen depletion can increase tumor hypoxia and activate hypoxia-activated therapy. Gluconic acid generation can enhance the acidity of the tumor microenvironment and trigger pH-responsive drug release. $H_2O_2$ can be converted into toxic hydroxyl radicals through the Fenton reaction and kill cancer cells. To enable GER to occur in the antenna's near-field, GOD was immobilized on the antenna, and glucose was passed through the microfluidic channel to react with GOD.

Initially, the chips were subjected to immersion in an acetone solution for 30 min at room temperature to eliminate impurities, followed by washing three times with deionized water and IPA, and drying with nitrogen. Subsequently, the surface of the clear chip was treated with oxygen plasma to generate hydroxyl groups. Following the hydroxylation process, the chips were immersed in a GO dispersion (1 mg/mL, 50 °C) to immobilize GO on the chip surface via covalent bonding, where the GO dispersion was ultrasonically pretreated. Then, the chemical linking of GO and GOD was achieved by incubating the chip in a cross-linker solution of 20 mM 1-ethyl-3-(3-dimethylaminopropyl) carbodiimide (EDC) and 40 mM N-hydroxysuccinimide (NHS) for 60 min. EDC is a zero-length crosslinker that activates the carboxyl groups on the surface of GO, while NHS reacts with the EDC-activated carboxyl groups to form stable amide bonds with the amino groups on the enzyme surface. By mixing EDC and NHS with GO and GOD, respectively, a chemical bond can be formed between the two materials, thereby immobilizing GOD on the surface of GO. Notably, the pH of the reaction mixture plays a critical role in the cross-linking process, affecting the activation degree of carboxyl groups on the graphene oxide surface and the reactivity of NHS molecules towards activated carboxyl groups. The pH was adjusted to 6.1 using MES buffer (2-(N-Morpholino) ethanesulfonic acid, pH 5.5). Subsequently, the treated chips were immersed in coupling buffer (1 mg/mL GOD in PBS buffer) for 90 min to immobilize GOD on GO, with the pH set to 7.2 using PBS buffer (phosphate-buffered saline, pH 7.4). Finally, the functionalized chip was thoroughly washed with PBS buffer solution and deionized water and dried at 4 °C.

### Materials and apparatus

Glucose oxidase from Aspergillus niger, graphene oxide, 1-ethyl-3-(3-dimethylaminopropyl) carbodiimide, N-hydroxysuccinimide, 2-(N-Morpholino) ethanesulfonic acid, phosphate-buffered saline, and glucose were purchased from Sigma-Aldrich (Singapore). The field emission scanning electron microscope (Hitachi Regulus 8230) was employed for conducting SEM and EDX analysis. For thickness analysis, a commercial atomic force microscopy (Dimension Icon, Bruker Inc) was used. Infrared spectral measurements were taken using an FTIR spectrometer (Cary 660, Agilent Technologies) with an infrared

microscope (Cary 610, Agilent Technologies). Electron beam lithography (Raith GmbH) was utilized for fabricating the pattern of nanoantennas. The metal deposition process was carried out with a UHV electron-beam evaporation (ATC-T Series, AJA Int.).

## Data availability

All data generated in this study are provided in the Supplementary Information and Source Data file. Previously published structures from the Protein Data Bank (PDB) can be found under accession code 1CF3 [https://doi.org/10.2210/pdb1CF3/pdb] (glucose oxidase). Additional data related to this paper may also be requested from the corresponding author X.M. or C.L. upon request. Source data are provided with this paper.

## Code availability

Codes for simulation and theoretical calculation in this study are available from https://doi.org/10.5281/zenodo.10030866. Additional codes that support the findings of this study are also available from the corresponding author X.M. or C.L. upon request.

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

## Acknowledgements

This work was supported by the Advanced Research and Technology Innovation Center (ARTIC) under the research grant of R-261-518-009-720/A-0005947-20-00; Ministry of Education (MOE) under the research grant of R-263-000-F18-112/A-0009520-01-00 at National University of Singapore.

## Author contributions

H.Z. and C.L. conceived the project. H.Z. performed antenna design, fabrication, testing, and deep learning algorithms. D.L. performed simulations. H.Z., Z.R., C.X., X.M., and C.L. discussed and analyzed the numerical and experimental results. C.L. supervised the project. All authors prepared the manuscript.

## Competing interests

The authors declare no competing interests.
