## [Peer Review File · Nature Communications]

Dynamic Construction of Refractive Index-Dependent Vibrations Using Surface Plasmon-Phonon PolaritonsEditorial Note: This manuscript has been previously reviewed at another journal that is not operating a transparent peer review scheme. This document only contains reviewer comments and rebuttal letters for versions considered at *Nature Communications*.

REVIEWERS' COMMENTS

Reviewer #1 (Remarks to the Author):

The authors worked diligently to prepare a very detailed response to my comments as well as to the comments of the other reviewers, and many of the technical comments have been addressed. However, the authors continue to overstate the novelty of their results. Some examples below (all taken from the blue text added/modified in the revision):

Abstract: "we develop a distinctive vibration through the excitation of coupled surface plasmon-phonon polaritons." The authors do not develop/invent a new type of resonance, they merely combine resonant functionality from two systems. The current wording gives the impression that this is a breakthrough, which it is not.

Abstract: "advance the frontier of biospectroscopy within biomedicine applications". Again, the simple glucose detection experiment shown in the manuscript is insufficient to make these bold claims (i.e., to revolutionize biomedical sensing).

Introduction, page 3: "However, the SEIRA enhancement is general without selectivity to molecular vibrations, rendering it unable to disentangle overlapping vibrational modes." Again, the core feature of SEIRA is its molecular selectivity! One may argue that SEIRAS struggles for some overlapping bands, but in general, this can also be resolved in other ways than the one proposed by the authors. As the authors state themselves, this problem is essentially solved in the literature: "Furthermore, the utilization of sophisticated data processing techniques, such as derivative spectroscopy²⁴, deconvolution²⁵, or artificial intelligence algorithms¹⁴, can contribute to discerning overlapping vibrational modes."

Discussion, page 18: "sensitivity mechanism of these unique vibrations" No new vibrations are invented.

I strongly suggest that the authors go through the whole manuscript again and carefully reword their claims of novelty to be more realistic.

Otherwise, on a technical standpoint, I think the manuscript can be accepted for publication in *Nature Communications*.

Reviewer #3 (Remarks to the Author):

The authors have provided thorough and extensive responses, addressing the questions and comments raised by all reviewers. It's evident that the authors have put in considerable effort to address these concerns, and I appreciate their diligence in this regard. Specifically, the authors have decided to take one subject out of this manuscript and to transfer some of the content to the supplementary material. Also, they have invested efforts to enhance the manuscript's clarity and coherence to make it more accessible to a broader readership.

However, I still am concerned that the manuscript appears excessively long and detailed. While I understand the importance of providing a comprehensive account of their work, it is essential to balance providing sufficient information and overwhelming the reader with excessive details. Also, even the authors agree that the performance improvements discussed in the manuscript are incremental. The authors have narrowed down their main contributions to focus on two things: studying RI-dependent vibrations and improving the identification of overlapping vibrational modes. While these are interesting and have some novelty, they are still limited in scope and for general use applications.

I am debating whether additional revisions to make the paper more concise and improve its overall readability can address my concerns adequately. It might be beneficial to consider seeking a fresh review from a new reviewer who can evaluate the manuscript impartially and help determine its suitability for publication in Nature Comm.

We express our profound gratitude to the reviewers for diligently scrutinizing our manuscript and providing valuable constructive comments. Incorporating their insightful comments has significantly enhanced the quality of our manuscript. Please find our point-by-point response below.

1. Response to Comments from Reviewer #1

The authors worked diligently to prepare a very detailed response to my comments as well as to the comments of the other reviewers, and many of the technical comments have been addressed. However, the authors continue to overstate the novelty of their results. Some examples below (all taken from the blue text added/modified in the revision):

Abstract: “we develop a distinctive vibration through the excitation of coupled surface plasmon-phonon polaritons.” The authors do not develop/invent a new type of resonance, they merely combine resonant functionality from two systems. The current wording gives the impression that this is a breakthrough, which it is not.

Abstract: “advance the frontier of biospectroscopy within biomedicine applications”. Again, the simple glucose detection experiment shown in the manuscript is insufficient to make these bold claims (i.e., to revolutionize biomedical sensing).

Introduction, page 3: “However, the SEIRA enhancement is general without selectivity to molecular vibrations, rendering it unable to disentangle overlapping vibrational modes.” Again, the core feature of SEIRA is its molecular selectivity! One may argue that SEIRAS struggles for some overlapping bands, but in general, this can also be resolved in other ways than the one proposed by the authors. As the authors state themselves, this problem is essential solved in the literature: “Furthermore, the utilization of sophisticated data processing techniques, such as derivative spectroscopy²⁴, deconvolution²⁵, or artificial intelligence algorithms¹⁴, can contribute to discerning overlapping vibrational modes.”

Discussion, page 18: “sensitivity mechanism of these unique vibrations” No new vibrations are invented.

I strongly suggest that the authors go through the whole manuscript again and carefully reword their claims of novelty to be more realistic.

Otherwise, on a technical standpoint, I think the manuscript can be accepted for publication in Nature Communications.

RESPONSE: We sincerely thank the reviewer for the valuable comments and the suggestion of publication in Nature Communications. In response to the reviewer's suggestion, we meticulously revisited the entire manuscript, refining certain assertions to mitigate the risk of overstatement.

The examples highlighted by the reviewer have been accordingly revised as follows (marked in red):

- Abstract: “we develop a distinctive vibration through the excitation of coupled surface plasmon phonon polaritons. Here, we **investigate the coupled mode of localized surface plasmon and surface phonon polaritons for vibrational de-overlapping. The coupling mode** is sensitive to the molecular refractive index,”
- Abstract: “advance the frontier of biospectroscopy within biomedicine applications **provide a valuable toolkit for biomedicine applications**”.
- Introduction, page 3: “However, the SEIRA enhancement is general without selectivity to molecular vibrations, **rende ring it unable it is challenging for SEIRA** to disentangle overlapping vibrational modes.”
- Discussion, Line 341, page 12: “sensitivity mechanism of these **unique** vibrations”.

In addition to the aforementioned examples, we have double-checked the entire manuscript and revised potential overstatements, as detailed below:

- Lines 45-46: “The utilization of artificial intelligence in sensory nanoantennas also offers novel **valuable** perspectives in biomedical science and measurable advancements in clinical diagnostics.”
- Line 328: “The **distinctive** feature of the SP-PhP platform compared to the common nanorod platform is the presence of RI-dependent SPhP vibrations.”
- Line 338: “This **unique** vibration is a specific type of SPhP oscillation occurring

the reststrahlen band of polar dielectric crystals”

- Lines 330-331: “In contrast to previously reported infrared antenna techniques for biospectroscopic applications, our platform represents a **substantial** advancement in accurately resolving overlapping vibrational modes (Supplementary Table 3).”
- Line 338: “This vibration is a **specific** type of SPhP oscillation occurring within the reststrahlen band of polar dielectric crystals, exhibiting a notable sensitivity to the molecular refractive index.”
- Line 359: “Our decoupling technique **is particularly attractive** **holds promise** for biomedical applications such as molecular screening, protein conformation and dynamics resolving, and pharmaceutical analysis, owing to the presence of molecular massive overlapping vibrational modes in these applications.”
- Line 369: “Finally, our de-overlapping concept can **___ could** be extended to other reactions involving overlapping vibrations, like protein condensation reactions⁵⁸.” We have incorporated these revisions into the manuscript to address the reviewer’s concern.

2. Response to Comments from Reviewer #3

The authors have provided thorough and extensive responses, addressing the questions and comments raised by all reviewers. It's evident that the authors have put in considerable effort to address these concerns, and I appreciate their diligence in this regard. Specifically, the authors have decided to take one subject out of this manuscript and to transfer some of the content to the supplementary material. Also, they have invested efforts to enhance the manuscript's clarity and coherence to make it more accessible to a broader readership.

However, I still am concerned that the manuscript appears excessively long and detailed. While I understand the importance of providing a comprehensive account of their work, it is essential to balance providing sufficient information and overwhelming the reader with excessive details. Also, even the authors agree that the performance improvements discussed in the manuscript are incremental. The authors have narrowed down their main contributions to focus on two things: studying RI-dependent vibrations and improving the identification of overlapping vibrational modes. While these are interesting and have some novelty, they are still limited in scope and for general use applications.

I am debating whether additional revisions to make the paper more concise and improve its overall readability can address my concerns adequately. It might be beneficial to consider seeking a fresh review from a new reviewer who can evaluate the manuscript impartially and help determine its suitability for publication in Nature Comm.

RESPONSE: We appreciate the reviewers' recognition of our efforts in addressing their concerns.

Regarding the concern that “the manuscript appears excessively long and detailed”, it is noteworthy that “Nature Communications publishes original research in one format, Articles, which may range in length from short communications through to **more in-depth studies**”, as stated in the Guide to Authors. Our manuscript presents a kind of in-depth study of identifying overlapping molecular vibrations, encompassing methodology,

theoretical analysis, experimental validation, and practical application demonstration. The main text is about 4300 words. As suggested by the reviewer in the last comments, we have transferred some of the content to the supplementary material, aiming to strike a balance between providing sufficient information and ensuring readability. In this new version, we have double-checked the entire manuscript and relocated specific details to the Methods section. This restructuring enhances manuscript readability, allowing interested readers to find detailed information in the Methods section. For instance, the term definitions (Lines 432-433 of the manuscript) and DNN model structure (Lines 438-442 of the manuscript) are moved into the Methods section in the new version.

In terms of “limited in scope and for general use applications”, as we claimed in the Discussion section, our de-overlapping concept could be extended to other reactions involving overlapping vibrations, like protein condensation reactions (discussed in Lines 366-367 of the manuscript). It is attractive as almost all biological reactions involve changes in molecular overlap. Hopefully it solves the reviewer’s concern.

Revision in the manuscript (content moved to Methods section):

- (Lines 432-433, manuscript): “The figure of merit (FOM) is defined as Sensitivity/FWHM, where FWHM is full-width at half maximum.”
- (Lines 438-442, Manuscript): “The DNN model consists of input, hidden, and output layers. The input and output layers comprise 1714 and 5 nodes, respectively, matching the spectral wavenumber points and reactants’ vibrational numbers. The model comprises two hidden layers, each consisting of 20 nodes, dedicated to extracting high-level features from the input data. This design facilitates the learning of intricate relationships, leading to accurate predictions.”